# Nitric Oxide, Nitric Oxide Formers and Their Physiological Impacts in Bacteria

**DOI:** 10.3390/ijms231810778

**Published:** 2022-09-15

**Authors:** Jinghua Chen, Lulu Liu, Weiwei Wang, Haichun Gao

**Affiliations:** Institute of Microbiology, College of Life Sciences, Zhejiang University, Hangzhou 310058, China

**Keywords:** nitric oxide, nitric oxide forming enzymes, nitric oxide synthase, hemoproteins, NO signaling, NO tolerance

## Abstract

Nitric oxide (NO) is an active and critical nitrogen oxide in the microbe-driven nitrogen biogeochemical cycle, and is of great interest to medicine and the biological sciences. As a gas molecule prior to oxygen, NO respiration represents an early form of energy generation via various reactions in prokaryotes. Major enzymes for endogenous NO formation known to date include two types of nitrite reductases in denitrification, hydroxylamine oxidoreductase in ammonia oxidation, and NO synthases (NOSs). While the former two play critical roles in shaping electron transport pathways in bacteria, NOSs are intracellular enzymes catalyzing metabolism of certain amino acids and have been extensively studied in mammals. NO interacts with numerous cellular targets, most of which are redox-active proteins. Doing so, NO plays harmful and beneficial roles by affecting diverse biological processes within bacterial physiology. Here, we discuss recent advances in the field, including NO-forming enzymes, the molecular mechanisms by which these enzymes function, physiological roles of bacterial NOSs, and regulation of NO homeostasis in bacteria.

## 1. Introduction

Nitric oxide (NO) is a reactive and highly diffusible gaseous molecule, which plays various physiological roles within bacterial and mammalian cells. The best-known function of NO in mammals is as a signaling molecule, impacting diverse physiological processes including vasodilation, neurotransmission, immunity, apoptosis, reproduction, and metabolism [1]. The significance of some early findings is evident; pioneer scientists in the field, Robert F. Furchgott, Louis J. Ignarro, and Ferid Murad were awarded the 1999 Noble Prize in Physiology or Medicine. In fact, inhibition of bacterial pathogen growth by NO, which for centuries has also underpinned nitrite (NO_2_^−^) application in meat product preservation, had revealed long before [2,3,4]. This effect later was later identified as apparent in mammalian host defense cells against a variety of pathogens and some viruses [5,6,7].

Nitric oxide synthase (NOS), a family of enzymes that catalyze NO production through oxidation of L-arginine (Arg) to L-citrulline (Cit), is the only source of endogenous NO in mammals [1]. Mammalian NOS (mNOS) exists in three isoforms that are responsible for the regulated synthesis of NO in distinct localizations, endothelial NOS (eNOS), inducible NOS (iNOS), and neuronal NOS (nNOS) [8]. NOSs from the plant kingdom have also been identified, but only in a few algal species such as green algae *Ostreococcus tauri* [9]. Seemingly, NOSs are not conserved in land plants, although the NOS-like enzyme AtNOS1 of *Arabidopsis thaliana* was reported nearly 20 years ago [10,11]. In bacteria, NO can be produced via two routes: respiratory and non-respiratory. The respiratory route consists of key reactions in transformation of nitrogen compounds, especially nitrogen oxides (NOx) which could serve as electron acceptors (EAs) for respiration, as well as reduction of NO_2_^−^ to NO by heme- or Cu-containing nitrite reductases during the denitrification process, and oxidation of ammonia (NH_3_) to NO by hydroxylamine (NH_2_OH) oxidoreductase (HAO) [12,13,14] (Figure 1). These enzymes along with others in the same pathways have a profound impact on the global nitrogen biogeochemical cycle [15].

The non-respiratory route is composed exclusively of bacterial NOS (bNOS), which are phylogenetically homologous to mNOSs and represent a simpler form from which eukaryotic NOSs evolve [16]. Central to the physiological activity of endogenous NO derived from bNOS activity is the radical-species nature of NO [17]. Not surprisingly, numerous studies have concluded that NO, either supplemented exogenously or generated endogenously by bNOS, acts as a bacteriostatic agent by damaging many cellular proteins carrying redox centers, such as heme, Fe-S clusters, and thiol groups [18]. Despite this, physiological impacts of bNOS-generated NO on bacterial cells are often species-specific, such as *Deinococcus radiodurans* NOS for UV damage recovery, *Silicibacter* NOS for signaling biofilm formation, *Bacillus subtilis* NOS for protection from oxidative stress, *Bacillus anthracis* NOS for aiding pathogen virulence, and *Staphylococcus aureus* NOS for regulating oxygen (O_2_)-based respiration, to name a few [19,20,21,22,23,24,25]. Recently, NO has been found to play an important role in governing communal bacterial behaviors, including formation and dispersion of biofilms, and in the biosynthesis of secondary metabolites [26,27,28].

Given the profound biological roles of NO in mammalian as well as bacterial cells, control of NO homeostasis is critical for therapeutic treatments for NO-related diseases, for maintaining a balanced nitrogen biogeochemical cycle, and for development of NO-dependent biotechnologies. NO production can be regulated in at least two ways, by selectively inhibiting the activity of NO-forming enzymes, NOSs in particular, and by mediating expression of their coding genes [1,15]. In this review, we discuss the origin of NO and bacterial NO-forming enzymes, functioning mechanisms of bNOS, and their biological impacts, and we highlight recent progress in areas of bNOS research, to expand the understanding of NO biology in bacteria.

## 2. Emergence of NO in the Earth’s Atmosphere and NO-Forming Enzymes

NO, a diatomic uncharged gas radical with one unpaired electron, is probably one of the earliest oxidants to have formed abiotically from the prebiotic atmosphere on earth [29,30,31]. Because of the reducing nature of the prebiotic environment, newly formed NO would be rapidly converted to nitroxyl (HNO), which entered the oceans due to its high water solubility [32]. In aqueous environments, multiple photochemical reactions involving HNO are known to produce various nitrogen intermediates, such as NO_2_^−^ and nitrate (NO_3_^−^), eventually leading to formation of stable NOx [33]. During a long history of evolution, living organisms emerged, interacted with nitrogen species, developed the ability to produce NO biologically, and shaped the ecosystem we see today (Figure 1). At present, NO is regarded as one of the central molecules linking NO_3_^−^ in both directions to dinitrogen gas (N_2_) and NH_3_, as part of the global nitrogen biogeochemical cycle [14,15].

NO generated biotically has been suggested as an intermediate of the route through which microbes gained ability to utilize nitrogen species that were abundant in the oceans, such as NH_4_^+^, NO_2_^−^, and NO_3_^−^, as sources of nitrogen for survival and growth [34]. NO can be released from nitrite reduction and hydroxylamine oxidation, reactions in denitrification and nitrification respectively [35,36] (Figure 1). In the present, retention of these abilities is restricted mainly to prokaryotes, which serve as the predominant force that drives the global nitrogen cycle [15]. In the case of bNOS, however, it remains unclear when and why this type of NO-forming enzyme in prokaryotes emerged. It has been suggested that bNOS may play a critical role in protecting ancestral microorganisms from various stresses, especially stress associated with reactive ozone (O_3_) [29]. In addition, recent studies have revealed the physiological significance of bNOS in respiration, suggesting that bNOS may have a more profound role in mediating bacterial physiology than was previously thought [23,24]. Despite this, it is also clear that a considerable portion of prokaryotes lack bNOS, although homologues of NOS have been identified in all kingdoms of life, implying that certain specific conditions may be required for this type of NO-forming enzyme to evolve [37,38].

## 3. NO Sources in Bacteria

### 3.1. NO Formation in the Respiratory Route

Bacteria are renowned for metabolic diversity and this feature is reflected in biotransformation of NOx, including NO. While a portion of bacteria are not NO formers, many generate NO by respiratory and/or non-respiratory routes, depending on species and even strains [18,39]. A variety of enzymes are involved in denitrification, dissimilatory nitrate reduction, and ammonia oxidation, which constitute the respiratory route and catalyze oxidoreduction reactions to release NO in prokaryotes [4,40,41] (Figure 1). These include NO-forming nitrite reductases (NirS and NirK), cytochrome *c* nitrite reductase (NrfA), and nitrate reductase NarGHI [42,43]. NirK/NirS, key enzymes of the bacterial denitrification pathway, are bone fide NO producers that carry out the one-electron reduction of NO_2_^−^ to NO in denitrifying bacteria (Figure 1). In contrast, NrfA and NarGHI may act as moonlighting NO producers because they primarily catalyze reduction of NO_2_^−^ and NO_3_^−^ to NH_3_ and NO_2_^−^, respectively [15]. When coupled with nitrate reductases, such as membrane-bound NAR and periplasmic NAP, the NO-forming nitrite reductase is able to generate NO starting with NO_3_^−^ [41].

NirS, also called cytochrome *cd*_1_ nitrite reductase (*cd*_1_-type), and NirK, copper-containing nitrite reductase (Cu-type), are structurally different but functionally equivalent [41]. NirSs, well studied in α-proteobacterium *Paracoccus denitrificans* and γ-proteobacterium *Pseudomonas aeruginosa*, are ~120 kDa homodimers with each subunit containing one heme *c* and one unusual heme *d*_1_ [44,45]. NirSs are highly conserved and display high homogeneity in structure. Despite this, because of explosive growth in genome sequences, comparative genomics have suggested that atypical *cd*_1_-type enzymes may exist. For example, *Pyrobacullum aerophilum*, a hyperthemophilic denitrifying crenarchaeon, encodes a protein that shares considerable sequence similarity to characterized NirSs while lacking heme *c* [46].

Compared to NirS, NirKs are more widespread in gram-positive and gram-negative bacteria as well as in Archaea [45,47]. The typical NirK enzymes exist as homotrimers with each monomer (~40 kDa) containing two distinct Cu-centers (Cu^I^ and Cu^II^) [47,48]. Atypical Cu-type enzymes have been also reported. β-proteobacterium *Ralstonia pickettii* NirK carries an additional heme-binding domain within each monomer [49], and *Thermus scotoductus*, a bacterium of the *Deinococci* class, possesses a NirK with three copper centers per monomer instead of two [50]. In contrast to atypical NirSs deduced from the sequence, the atypical NirK enzymes are functionally substantiated [49,50].

In recent years, HAO has been found to be an NO former in ammonia-oxidizing bacteria (AOB) during nitrification [12,36] (Figure 1). HAO, a cytochrome *c* which was originally reported to oxidize NH_2_OH to NO_2_^−^ in *Nitrosomonas* [51,52], catalyzes an essential step of the oxidation of NH_3_ to NO_3_^−^ [15,53]. With the heme P460 cofactor as the site of catalysis, HAO oxidizes NH_2_OH to NO under anaerobic or aerobic conditions; incontrast, NO production was regarded as a result of incomplete NH_2_OH oxidation in some early studies [36,54].

### 3.2. NO Formation in the Non-Respiratory Route

Generation of NO through non-respiratory routes in bacteria is carried out by bNOS [41]. To date, bNOS enzymes have been found in many bacterial species and strains, including *B. subtilis*, *D. radiodurans*, *Geobacillus stearothermophilus*, *Lactobacillus fermentum*, *Nocardia*, *S. aureus*, *Sorangium cellulosum*, *Streptomyces*, *Synechococcus* PCC 7335, and even in archaeon *Natronomonas pharaonis* [55,56,57,58,59,60,61,62,63,64,65]. Although bNOS shares over 40% amino-acid sequence identity with the oxygenase domain of mNOS, they differ from each other drastically in protein domain organization [37] (Figure 2). mNOS is composed of an isozyme-specific N-terminal domain, a β-hairpin hook, a Cys-X_4_-Cys motif that coordinates a zinc atom, an oxygenase domain, a calmodulin-binding site, and a reductase domain [17]. In contrast, bNOSs tend to be single-domains proteins of the oxygenase domain, although exceptions exist [9,16,62] (Figure 2). Despite this, bNOS, catalyzes 5-electron oxidation of Arg to produce Cit and NO using cellular reducing equivalents, the same way as its mammal counterpart [16]. The reaction consists of two independent steps to generate NO; first, Arg is oxidized at its terminal (omega) guanidine nitrogen, resulting in the intermediate N(omega)-hydroxy-L-arginine (N-OH-L-Arg); N-OH-L-Arg is the substrate for the second step during which it is oxidized to release the terminal guanidine nitrogen as NO, with the generation of Cit [37].

## 4. Structure and Function of NO-Forming Enzymes in Bacteria

### 4.1. NO-Forming Enzymes in the Respiratory Route

Nitrite reductases (NiRs) are a group of enzymes that catalyze either one-electron reduction of NO_2_^−^ to NO during denitrification, or six-electron reduction of NO_2_^−^ to NH_3_ during ammonification [47,66]. To date, the structures of both NirSs and NirKs have been determined and it is interesting to find that these two enzymes are structurally different but functionally equivalent.

NirSs are homodimers with each subunit containing an α-helix enriched heme *c* binding domain and a unique heme *d*_1_ binding β-propeller domain [17,41] (Figure 3A). The former functions as electron acceptor (EA) and the latter works as the catalytic center. Previous studies of NirSs from *Paracoccus pantotrophus* and *P. aeruginosa* demonstrated that although these two NirSs exhibited similar overall structures, significant conformational changes of the heme coordination resulting from the reduction of heme *d*_1_ were observed in oxidized and reduced states [44,67,68] (Figure 3B,C). In addition, it should be noted that NirSs are bifunctional enzymes that catalyze not only the one-electron reduction of NO_2_^−^ to nitric oxide, but also the four-electron reduction of O_2_ to water (H_2_O) as part of the respiration process.

NirKs are homotrimers with each subunit harboring two distinct copper centers; one (Cu-I) is responsible for the electron transfer and the other (Cu-II) works as the catalytic site. The Cu-I is located at the interior of each monomer while the Cu-II resides at the interface between monomers, causing the copper ions to experience different local environments [17] (Figure 4A). The copper ion of Cu-I is coordinated by two histidines, one cysteine and one methionine, while the copper ion of Cu-II is coordinated by three histidines and a solvent molecule [34] (Figure 4A). In addition, two atypical NirKs isolated from *Thermus scotoductus* and *Ralstonia pickettii* were found to possess one extra copper center (Cu-I) and one additional heme-containing domain, respectively [35,36] (Figure 4B,C).

Multiheme HAOs also play important roles in aerobic and anaerobic ammonia oxidizers, such as ammonia-oxidizing bacteria (AOB) and methanotrophs, by converting NH_2_OH to NO. Within AOB, both NH_2_OH and NO are obligate intermediates during NH_3_ oxidation [36]. HAO oxidizes NH_2_OH to NO by a three-electron reaction under anaerobic or aerobic conditions. In contrast, NO_2_^–^ produced in HAO activity assay is a nonenzymatic product resulting from the oxidation of NO by O_2_ under aerobic conditions [39,40]. Interestingly, structural studies have demonstrated that the NO-forming octaheme HAO (KsHAO) from anammox bacterium *Kuenenia stuttgartiensis* shares several characteristics with the octaheme nitrite-forming HAO from *Nitrosomonas europaea* (NeHAO). Despite the lower sequence identity, the overall structures of KsHAO (PDB ID: 4N4J) and NeHAO (PDB ID: 4N4N) are very similar, including the interior heme arrangements [52,69]. The heme configuration of KsHAO and NeHAO partially resembles that of pentaheme nitrite reductases (NrfAs), in that the five hemes in NrfAs perfectly superimpose onto hemes 4–8 of KsHAO and NeHAO [42,69]. However, in the trimeric KsHAO and NeHAO, hemes are arranged in a closed ring. The only significant structural difference between the active sites of KsHAO and NeHAO is that a tyrosine (Tyr358) at the distal side of the porphyrin ring within NeHAO is replaced by a hydrophobic methionine (Met323) in KsHAO; however, with this subtle change the question is retained of what determines the reaction specificity of HAOs. In aerobic and nitrite-dependent methanotrophs, NH_2_OH is a toxic intermediate which needs to be rapidly eliminated. The HAO from the thermophilic verrucomicrobial methanotroph *Methylacidiphilum fumariolicum* SolV possesses a characteristic P_460_ chromophore and catalyzes the rapid oxidation of NH_2_OH to NO [70].

### 4.2. bNOSs

mNOSs are biochemically and structurally characterized prior to bNOSs. All three mNOS isoforms are homo-dimeric, with each monomer comprising six conserved domains, namely N-terminal isozyme-specific domain, Cys-X_4_-Cys zinc-binding site, NOS oxygenase (NOS_oxy_) domain, β-hairpin hook, calmodulin-binding site, and C-terminal flavoprotein reductase (NOS_red_) domain (Figure 2). The NOS_oxy_ domain of mNOS harbors cofactors of heme and 6R-tetrahydrobiopterin (BH_4_), and functions as the substrate (Arg)-binding site. The mNOS_red_ domain contains the binding sites for FAD, FMN, and NADPH [17].

To date, the structures of three different bNOSs isolated from *B. subtilis*, *S. aureus*, and *G. stearothermophilus* have been resolved [57,71]. Most bNOSs possess only the oxygenase domainand lack the remaining counterpart domains in mNOSs (Figure 2). The overall architecture of bNOSs exhibits a similar organization to the extensively studied oxygenase domain of mNOSs [8]. Commonly, each NOS monomer contains a winged β-sheet core which resembles a baseball catcher’s mitt and binds a heme molecule [8,61] (Figure 5A). However, the cofactor BH_4_ binding in mNOSs is substituted by tetrahydrofolate (FH_4_) in bNOSs [16].

Dimerization is necessary for the catalytic activity of all NOSs in vitro, and possibly regulates activity in vivo [8,72]. Intriguingly, between bNOSs and mNOSs there exist many differences of interactions at the interface. Four conserved surface regions together with the N-terminal hook and the Cys-X_4_-Cys zinc-binding site are responsible for stabilizing the mNOS dimer [17,73]. Nevertheless, lacking the modules of β-hairpin hook and the Cys-X_4_-Cys site, bNOSs make several compensatory structural changes including the mutual subunit π-π stacking resulting from aromatic residues, the increased interface hydrophobicity by a conserved proline residue, and multiple hydrogen bonds and van der Waals interactions [8,17,71] (Figure 5A,B). Substrate and cofactor binding have been found to strengthen the dimerization of bNOSs as well as mNOSs [1]. Both of the pterins, BH_4_ in mNOS and FH_4_ in bNOS, are closely associated with the surrounding residues at the interface of the NOSoxy dimer; however, the detailed environments of pterin-binding sites within bNOS and mNOS are different [8,71].

The oxygenase domain of all NOSs contains a heme-containing catalytic pocket which provides the binding sites for substrate (Arg). Together with the adjacent pterin, a triangular association between heme, pterin, and Arg mainly via hydrogen bonds forms the catalysis reaction center for NO formation (Figure 5B). The proximal cysteine coordinates with the heme-iron and forms hydrogen bonds with a nearby tryptophan, and this Cys–Trp interaction is thought to increase the heme redox potential and to regulate the NOS-mediated catalysis process [74,75,76,77]. In mNOS_oxy_, a protruded hydrophobic ‘helical lariat’ binds one pterin (BH_4_), while bNOS conserves only a partial counterpart pterin-binding site. Instead, the pterin molecule in bNOS is stabilized by forming hydrogen bonds with the heme propionate D and substrate Arg, and by a π-π stacking interaction with a conserved Trp residue. Substrate Arg locates at the distal side of porphyrin ring, with its guanidino group close to the central iron atom and its amino/carboxyl groups forming hydrogen bonds with three residues (tyrosine, tryptophan, and glutamate) above. Additionally, a conserved Isoleucine adjacent to the substrate Arg in bNOS substitutes for a conserved counterpart valine which is related to efficient NO release in mNOS [76,77].

In addition to the canonical bNOSs mentioned above, in a photosynthetic cyanobacterium *Synechococcus* PCC 7335, one unique NOS or bNOS-like protein (SyNOS) was identified [63]. Differing from the canonical bNOSs, SyNOS contains one N-terminal globin domain in addition to the oxygenase and reductase domains, like that in mNOSs (Figure 2). It has been proposed that SyNOS participates in nitrogen assimilation from L-Arg in a tetrahydrobiopterin (H_4_B)-dependent manner, like other NOSs, and a high level (>200 μM) of Ca^2+^ is required for its activation even though it does not contain a Ca^2+^-calmodulin [25]. Interestingly, the globin domain of SyNOS functions to convert NO to NO_3_^−^ in the presence of O_2_ and NADPH, making SyNOS an enzyme having both NOS and NO oxygenase activities [25].

Some specific bNOSs have been found to participate in regulation of biosynthesis for some bacterial natural products (NPs). During the synthetic processes of thaxtomin A and rufomycin, two bNOSs (TxtD and RufN) provide NO molecules for the NO-dependent nitration reaction, catalyzed by cytochrome P450 homologues (TxtE and RufO), respectively [28,55,78,79,80]. Another bNOS (PtnF) was reported to involve an enzymatic and non-enzymatic cascade which caused the formation of a rare 1,2,3-triazolopyrimidine scaffold [55]. Although involved in the biosynthesis of different natural products, these special bNOSs exhibit similar spatial organizations and highly conserved catalytic centers like those of typical bNOSs, according the structures in the AlphaFold database [81,82] (Figure 6).

Due to the highly conserved catalytic site, the working modes of bNOSs and mNOSs are quite similar. The catalysis of Arg to NO and Cit is carried out in two successive stages. In the first stage, substrate Arg is hydroxylated to N^ω^-hydroxy-Larginine (NOHA), which in the second stage is further oxidized to Cit and NO. The detailed mechanistic and kinetic processes of bNOSs and mNOSs have been well studied and reviewed [16,17,83,84]. In brief, the ferric-heme (Fe^3+^) of bNOS initially accepts one electron from a reductase, causing the ferric iron to change to ferrous (Fe^2+^) state, and subsequently one O_2_ binds at the NOS catalytic site to form a highly reactive heme-based oxidative complex (Fe^2+^O_2_ or Fe^3+^O_2_^−^) [85]. A second electron from a pterin is then donated to the ferric superoxide species (Fe^3+^O_2_^−^) followed by uptake of one proton, which results in a heterolytic cleavage of the O-O bond within ferric peroxo (Fe^3+^O_2_^2−^) and the production of one NOHA and one H_2_O molecule. In the second stage, an NOHA-Fe^3+^O_2_^−^ intermediate product is first produced following one O_2_ binding to the ferric heme. After accepting a second electron from one pterin molecule, the process differs from the first stage; the heterolytic cleavage of the O-O bond within ferric peroxo (Fe^3+^O_2_^2−^) is coupled with a nucleophilic attack on NOHA, which results in the nitroxyl of NOHA bonded to a ferric-hydroxide intermediate and the formation of one Cit. Finally, one electron is fed back to the pterin radical with the formation of NO and the release of one H_2_O molecule.

## 5. Physiological Roles of NO in Bacteria

### 5.1. Inhibition of Growth

NO biology has rapidly become a fascinating area in the field of microbiology, since this gaseous molecule was found not only to constitute a defense line against bacterial infection when released by eukaryotic cells but also to underpin bacteriostatic effects of NO_2_^−^ [3,86,87]. Because inhibition of growth by NO lies at the center of bacterial NO physiology, many attempts have been made aiming at high-throughput screening for bacterial proteins susceptible to NO. A broad array of enzymes has been identified as potential targets of NO, and most of them contain redox centers [40,88]. Active redox enzymes identified to be primarily responsible for growth arrest upon NO exposure include enzymes containing cytoplasmic Fe-S, that can form dinitrosyl-iron complex (DNIC) with NO, such as aconitase, argininosuccinate synthase, fructose-1,6-biphosphate aldolase, pyruvate dehydrogenase, and α-ketoglutarate dehydrogenase [89,90,91,92,93]. Also involved are heme-containing metabolic enzymes outside the cytoplasm, heme-copper oxidases, and cyt *bd* oxidases in particular, that form ferrous-nitrosyl complex, and some thiol proteins, such as lipoamide dehydrogenase LpdA, that form S-nitrosothiols [94,95,96,97,98] (Figure 7). It should be noted that NO released by bNOSs is unlikely to be evenly distributed and may reach relatively high concentrations in certain locations in the cell. Because of this proximity effect, NO may inhibit some but not all potential targets that are characterized by the same feature.

As a small uncharged gaseous molecule, NO can easily pass through the biological membrane and diffuse into the cytoplasm [99] (Figure 7), implying that exogenous NO at high concentrations can react with biomolecules both inside and outside the cytoplasm [100,101]. Identification of cyts *c* as a target of NO in bacteria further suggests that NO interacts with redox-active proteins, and more critically, NO appears to interact with these proteins indiscriminately [102] (Figure 7). Cyts *c* represent a large group of heme-containing proteins located exclusively outside the cytoplasm (either membrane-bound or soluble in the periplasm of gram-negative bacteria), which are structurally featured with each heme covalently attached to the polypeptide [103,104]. In line with their hemoprotein nature, cyts *c* are highly susceptible to NO, according to in vitro biochemical analyses and in vivo physiological assessment [105,106]. In *Shewanella oneidensis*, a γ-proteobacterium renowned for respiratory versatility owing to its large repertoire of cyts *c* (up to 42), the loss of all these but not small quantities elicited a drastic difference in the sensitivity of cells to NO [107,108,109]. Consistently, the NO tolerance of *S. oneidensis* increased with the overall cyt *c* abundance, which can be manipulated by up-regulation of cyt *c* biosynthesis [102,110]. Importantly, once cyts *c* in *S. oneidensis* were depleted, cytoplasmic NO targets such as aconitase become hypersensitive to NO [102] (Figure 7). Therefore, in bacteria rich in cyts *c*, these hemoproteins tend to function as a major NO sink, resulting in reduced intracellular levels of free NO and thus protecting other growth-critical targets. In fact, the differences in physiological background among bacteria have been proposed to explain significant interspecies differences in NO targets associated with bacteriostasis, in some cases differing even within the same species [89,93].

Intriguingly, NO inhibition may play a beneficial role for pathogens. It has been found that NO generated from native bNOS plays an essential role in electron transfer and colonization of *S. aureus* [23]. The proposed mechanism is that under microaerobic conditions, NO targets heme-containing cytochrome oxidases to maintain membrane bioenergetics. Moreover, endogenous NO at appropriate levels has been found to improve growth and confer nitrosative stress tolerance in *Escherichia*
*coli* when produced by bNOSs of unicellular algae *O. tauri* and cyanobacterium *Synechococcus* PCC 7335 [111]. Two roles of NOS-generated NO have been proposed to explain these phenotypes; increasing Arg-metabolizing efficiency by functioning as a nitrogen source, and increasing viability of *E. coli* cells by reducing ROS production.

### 5.2. Interplay with Oxidative Stress

In addition to inhibiting growth as a bacteriostatic agent, NO, along with its derivatives including reactive nitrogen species (RNS) and reactive oxygen species (ROS), provides an arsenal to support a multifaceted pathogen eradication program, far more complex than direct killing by nitrosative and oxidative modification of critical bacterial macromolecules [18]. NO in isolation exerts little bactericidal activity against *E. coli*, but when combined with hydrogen peroxide (H_2_O_2_), it mediates a dramatic three-log increase in cytotoxicity [112]. This synergistic redox action can possibly be attributed to Iron-containing proteins, especially those with iron-sulfur clusters. Upon NO exposure, ferrous iron from the clusters is released into the interior of the cell, where it interacts with peroxide to form hydroxyl radical via Fenton chemistry, leading to the oxidation of DNA [112]. Additionally, NO as well as hydrogen sulfide (H_2_S) inhibits ROS scavengers, in particular catalases which are hemoproteins, allowing peroxide to kill the cell before being degraded [21,113,114]. Under these circumstances, the combination becomes extremely effective for eradicating bacterial cells [114].

Interestingly, the combinatorial redox action of NO and peroxide can generate contrasting effects, depending on the timing of application. When applied sequentially, NO played a role in protecting cells of *E. c**oli*, *Staphylococcus, and Bacillus* from ROS-mediated killing [21,115,116]. Similar phenomena have also been observed with the combination of H_2_S and H_2_O_2_ against a variety of bacteria [114,117]. The underlying mechanism is that inhibition of ROS scavengers by NO or H_2_S triggers cellular oxidative stress response, which causes drastically enhanced production of multiple ROS scavengers and damage-control proteins, and subsequently increases bacterial tolerance to ROS [114,118].

### 5.3. Biosynthesis of Natural Products

In recent years, a new understanding of the role of NO in biosynthesis of secondary metabolites in bacteria has been established [28]. Upon NO exposure, nitration (R-NO_2_) and nitrosation (R-NO) of biomolecules can both occur, resulting in inhibition of metabolic enzymes and DNA damage [86]. However, these processes can be repurposed by bacteria for synthesizing natural products. The biosynthesis of thaxtomin A relies on an NO-forming protein and a metalloenzyme catalyzing the NO-dependent chemistry, which are encoded by *txtD* and *txtE*, respectively, in the thaxtomin A biosynthetic gene cluster [60,78] (Figure 8). During the biosynthetic process, TxtD oxidizes Arg for production of pathway-dedicated NO, which provides a nitro group to be incorporated into L-Trp to form 4-nitrotryptophan, following a nitration reaction catalyzed by TxtE [28,78]. Although the precise mechanism remains to be fully elucidated, a similar strategy appears to be employed for biosynthesis of rufomycin, which relies on bNOS RufN and RufO that catalyze the NO-dependent chemistry [79,80] (Figure 8). Interestingly, this strategy does not apply to biosynthesis of 1,2,3-triazole [65] (Figure 8). Despite the non-enzymatic nature of the nitrosative cyclization reaction involving NO, the triazole biosynthesis system requires a pathway-specific bNOS, i.e., PtnF [65] (Figure 8). The role of NO in natural product biosynthesis stresses clearly the proximity effect of NO in bacteria. It should be noted that the current understanding of the NO-dependent biosynthesis of natural products is just the tip of the iceberg. A BLASTp search against TxtD returns over 500 results, implying that NO-dependent pathways are clearly underexplored [28]. In addition, there likely exist NO-dependent pathways that do not depend on a dedicated NO former, a strategy that has been suggested in the biosynthesis of pyrrolomycin and coenzyme F430 [71,119].

### 5.4. Regulation of Bacterial Communities

Although bacteria are generally regarded as archetypal unicellular organisms, in their natural environments they more frequently live in complex multicellular communities [120,121]. NO has been linked to formation of bacterial communities, regulating the spatial structures and emergent properties of these multicellular entities [27]. One well-studied example of such a community is the rhizobia–legume symbiosis, in which the role of NO differs from that known in pathogenic interactions [41]. During *Ensifer meliloti*–*Medicago truncatula* symbiosis, NO exhibits different or even opposite influences on the formation of nodules; while NO promotes symbiosis during the infection steps, in mature nodules it inhibits the expression and activity of nitrogenase and glutamine synthetase by tyrosine nitration, resulting in NH_4_^+^ assimilation [122,123,124,125]. In addition, recent investigations have revealed that NO serves as a signal for nodule senescence in nodules of *M. truncatula* and *Lotus japonicus* [126,127]. Moreover, between animals and bacteria NO is implicated in establishing symbiosis, for example, between *Euprymna scolopes* and *Vibrio fischeri* [128].

Biofilms are a common and widespread multicellular form of bacterial life in nature. Biofilms can form either on surfaces at the solid–liquid interface or at liquid–air interfaces, typically requiring an extracellular matrix composed of polysaccharides, proteins, and nucleic acids [120]. NO has been shown to affect biofilm formation and dispersal in a wide variety of gram-negative and gram-positive bacteria, a topic which has been well reviewed recently [18,27,129]. In *P. aeruginosa*, biofilm dispersal was induced by NO at very low concentrations, while a mutant lacking the only NO former did not disperse from biofilms, and a mutant deficient in NO scavenger exhibited enhanced dispersal [130,131]. Similarly, NO inhibited biofilm formation in *Vibrio fischeri* [132]. However, the effect of NO on *S. oneidensis* biofilm formation was positive although this bacterium is unable to produce NO endogenously [106,133]. In recent years, one of the most exciting achievements in uncovering the role of NO in regulation of biofilm formation and dispersal has been the identification of NO signaling cascades, composed of at least one protein that senses NO and one protein that regulates gene expression and/or enzyme activity in response to NO [27,134]. To date, most of the characterized bacterial NO sensors belong to H-NOX (heme-nitric oxide or O_2_ binding domain) proteins, which are homologous to the mammalian NO sensor, soluble guanylate cyclase [26,135,136]. Genes coding for NO-sensing H-NOX are frequently located next to those coding for its responding partners, such as cyclic di-GMP (c-di-GMP) synthases, or c-di-GMP phosphodiesterases, or histidine kinases of a two-component regulatory system [133,135]. Conceivably, NO commonly regulates biofilm formation in many bacteria by modulating intracellular c-di-GMP levels, and less frequently by interfering with quorum sensing or by forming a multi-component phosphorelay signaling cascade [137,138,139,140]. In addition to H-NOX, a novel family of hemoproteins known as NosP have been identified as NO sensors [20]. Similar to H-NOX, NosP homologues are widely distributed in bacteria and are encoded by genes in proximity with those for histidine kinases, or enzymes catalyzing c-di-GMP synthesis and degradation [129,141].

## 6. Regulation of NO Production in Bacteria

### 6.1. Regulation of NO Production in the Respiratory Route

Given the diverse physiological roles of NO in bacteria, it is conceivable that the homeostasis of NO must be carefully maintained. Regulation of NO production via the respiratory route has been extensively studied, and a relatively desirable understanding of the subject has been established [41]. However, the current understanding is mostly derived from investigations into production of nitrous oxide (N_2_O), the direct product of NO reduction and a potent greenhouse gas driving climate change [142]. Environmental factors and the redox state of the cell are critically involved in regulating the expression of key genes involved [143]; as respiration of non-oxygen EAs is substantially less energetic than O_2_, expression of genes for denitrification and dissimilatory NO_3_^−^ reduction is heavily repressed in the presence of O_2_ [143,144]. Expression of the *hao* gene may not be subjected to O_2_ repression, as NO could be generated by HAO under anerobic or aerobic conditions [36], although evidence is not available at present. In addition to O_2_, the key NOx in denitrification and dissimilatory NO_3_^−^ reduction, and environmental factors such as pH, and various metals (Mo, Fe, or Cu) that function as cofactors, have all been shown to influence expression of genes in these pathways [143,144,145,146,147]. The major regulators mediating responses to O_2_ and NOx belong to the cyclic AMP receptor protein (CRP)/fumarate and nitrate reductase regulation (FNR) superfamily, all referring to the same type of regulatory protein with similar domain structures. Example proteins of this superfamily whose roles in denitrification have been studied include DNR- and FNR-type regulators from *P. denitrificans*, *Rhodobacter sphaeroides*, and various *Pseudomonas* species [148,149,150].

A well understood example is the FixLJ-FixK2-NnrR regulatory cascade in *Bradyrhizobium diazoefficiens* (previously called *B. japonicum*), a model organism in the study of denitrification in rhizobia and the most widely used species in commercial inoculants for soybean crops [144,146,151,152]. FixK_2_, a CRP-FNR-type transcriptional regulator, directly controls expression of genes for reduction of NO_3_^−^ to NO_2_^−^ (*nap* genes), NO_2_^−^ to NO (*nirK*), and NO to N_2_O (*nos* genes) under microoxic conditions in the periplasm [153]. This protein is in fact a global regulator, mediating the expression of more than 300 genes, including not only those associated with microoxic metabolism (*fix* genes) and denitrification, but also several regulatory genes (*rpoN1*, *fixK1*, and *nnrR*) [153]. Subsequent response to NO is mediated by NnrR, another CRP-FNR-type transcriptional regulator directly controlling expression of the *noc* genes encoding cytochrome *c* NO reductase [154].

### 6.2. Regulation of NO Production in the Non-Respiratory Route

Many transcriptional factors involved in bacterial NO homeostasis with bNOS as the major NO source have been identified. Similar to those mediating NOx transformation in the respiratory route, these transcriptional regulators belong mainly to the CRP-FNR family [40]. However, the physiological impacts of these regulators predominantly effect the expression of genes that encode proteins conferring NO tolerance, and little is known about how they affect expression of bNOS genes [40].

Research into regulation of NO production by bNOSs has primarily focused on bNOS inhibitors, attempting to improve the effectiveness of antimicrobials, as endogenous NO has been shown to promote virulence and increase the tolerance of pathogens to oxidative stress [22,116,155]. The development of drugs targeting mNOS has been researched extensively for many years, and these studies not only established a framework combining crystallography, computational chemistry, and organic synthesis, but also developed many promising inhibitors, especially aminopyridine compounds, which could guide investigations into bNOS inhibitors [156]. One key challenge for the development of inhibitors with specificity towards bNOS over mNOS is that the active sites are nearly identical [157]. Despite this, some critical differences between bNOS and mNOS can be exploited for bNOS inhibitor design [157]. bNOSs contain only an oxygenase domain, but there exist some key amino acid variances between the NOS active sites [16]. For example, bNOS has an Asn residue that directly interacts with the l-Arg substrate, while this residue is Asp in most mNOSs [158]. Additional active site differences in bNOS include His128 (Ser in mNOS) and Ile218 (Val in mNOS). In addition, the difference between mNOS and bNOS at the pterin cofactor-binding site, which results in a thousand-fold difference in binding affinities of mNOS (at nM level) and bNOS (at μM level) for pterin, could be an attractive avenue [16,158]. Based on these differences, an array of compounds have been developed in recent years that specifically inhibit bNOS [159]. These include aminopyridine inhibitors and thiophenecarboximidamine inhibitors, both of which bind to both the active and pterin-binding sites [158,159,160,161,162].

## 7. Concluding Remarks

In this review, we have summarized recent advances in bacterial NO-forming enzymes and how they influence bacterial physiology. It is clear that enzymes respiring NOx as EAs prior to O_2_, including NO-forming nitrite reductases and HAO, are early contributors to energetics. These NO formers function exclusively outside the cytoplasm and play critical roles in shaping electron transport pathways in bacteria, and in driving the global nitrogen biogeochemical cycle. In contrast, bNOSs have evolved as intracellular enzymes catalyzing the metabolism of Arg. The inherent chemical reactivity of NO as a radical species challenges the development of all lifeforms, especially because of the proximity effect. Redox-sensitive proteins are primary targets of NO because of their susceptibility to modification by forming ferrous–nitrosyl and dinitrosyl–iron complexes as well as S-nitrosothiols. Despite this, it should be noted that cellular targets of NO responsible for growth inhibition tend to be different among bacteria, depending on species and even strains. Importantly, the proximity effect of NO may be essential to certain biological processes, such as NO-dependent biosynthesis of natural products, where coding clusters always contain a gene for a dedicated bNOS.

To date, various bacterial NO formers, including bNOSs, have been identified and characterized in terms of structure and biological function. It is interesting to find that despite certain differences in their interior cofactors, bNOSs largely resemble their counterpart mNOSs in aspects of spatial construction and catalytic properties. Most bNOSs only possess the oxygenase domain, and bind FH_4_-type of pterins instead of the BH_4_-type pterins in mNOSs. The heme-containing catalytic pocket that serves as the binding site for substrate (Arg) is highly conserved in all NOSs, and a close triangular association between heme, pterin, and Arg builds the catalytic reaction center for NO formation. To date, the functional structure of bNOSs in the catalysis of Arg to NO and Cit, similar to that of mNOSs, has been clarified to a large degree. However, studies of non-respiratory NO-forming enzymes, NiRs and multiheme HAOs, have revealed a different NO-producing mechanism, enriching the diversity of NO-forming systems in nature.

In this review, we briefly extended the discussion to consider the diverse biological processes on which NO may impact, and NO signaling in bacterial physiology. Bacterial cells have developed a sophisticated system to maintain NO homeostasis, many of which are transcriptionally responsive to NO. Finally, we considered bNOS inhibitors, which are now emerging as a promising drug for promoting the effectiveness of antimicrobials and beyond.

## Figures and Tables

**Figure 1 ijms-23-10778-f001:**
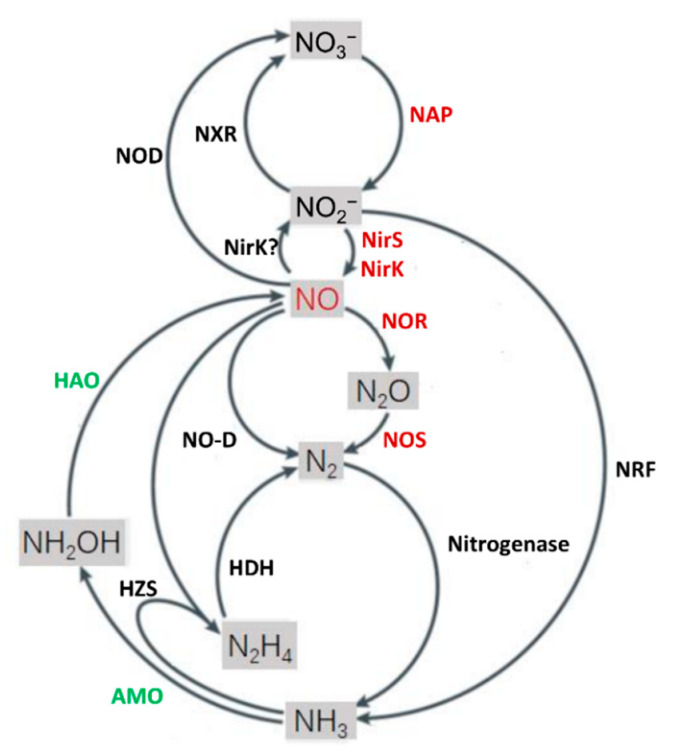
NO-centered transformation of inorganic nitrogen compounds in bacteria. Redox reactions include reduction of NO_3_^−^ to NO_2_^−^ by dissimilatory periplasmic nitrate reductase (NAP), reduction of NO_2_^−^ to NH_3_ by dissimilatory periplasmic cytochrome *c* nitrite reductase (NRF), reduction of NO_2_^−^ to NO by heme-containing (*cd*_1_-type, NirS) or copper-containing (Cu-type, NirK) nitrite reductases, oxidation of NO to NO_2_^−^ by NirK (?, this process is still under debate [15]), reduction of NO to N_2_O by nitric oxide reductase (NOR), reduction of N_2_O to N_2_ by nitrous oxide reductase (NOS), reduction of NO to N_2_ by nitric oxide dismutase (NO-D), oxidation of NO to NO_3_^−^ by nitric oxide oxidase (NOD), oxidation of NO_2_^−^ to NO_3_^−^ by nitrite oxidoreductase (NXR), oxidation of NH_3_ to NH_2_OH by ammonia monooxygenase (AMO), oxidation of NH_2_OH to NO by hydroxylamine oxidoreductase (HAO), reduction of N_2_ to NH_3_ by nitrogenase, reduction of NO to NH_2_OH and subsequent condensation with NH_3_ to form N_2_H_4_ by hydrazine synthase (HZS), and oxidation of N_2_H_4_ to N_2_ by hydrazine dehydrogenase (HDH). Denitrification and ammonia oxidation pathways that generate NO are in red and green, respectively.

**Figure 2 ijms-23-10778-f002:**
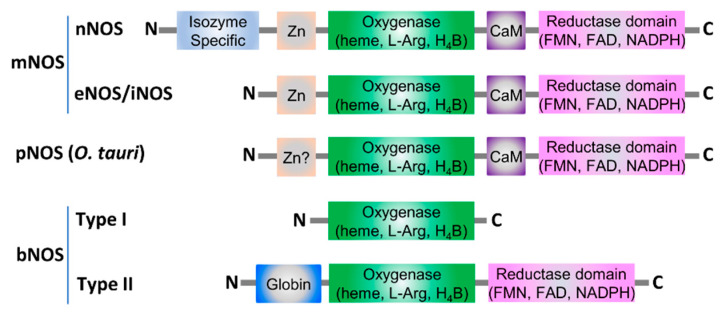
Structural organizations of mNOS, plant NOS (pNOS), and bNOS. mNOSs exists in three isoforms, nNOS, eNOS and iNOS. pNOSs have only been found in green algae species, as in *O. tauri*. bNOSs can be grouped into two types; a large majority of bNOSs belong to type I, which is composed of only one oxygenase domain homologous to that of mNOS. Type II bNOSs are rather rare, found only in a few species, that contain an N-terminal globin domain in addition to oxygenase and reductase domains, as seen in photosynthetic cyanobacterium *Synechococcus* PCC 7335.

**Figure 3 ijms-23-10778-f003:**
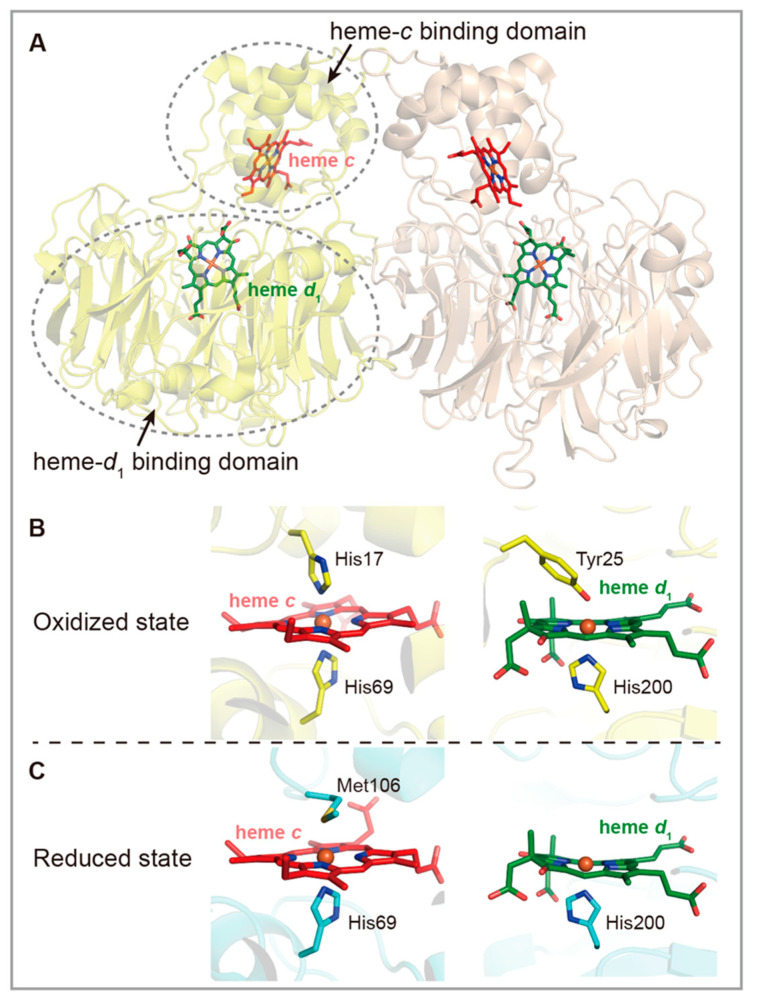
Structures of three copper-containing nitrite reductases (NirKs), and the coordination of copper ions and heme within each subunit. (**A**) The typical NirKs are homo-trimers with each monomer containing two different copper centers (Cu-I and Cu-II) (PDB ID: 1AS7). (**B**) The NOS from *Thermus scotoductus* possesses one additional I-type copper center (PDB ID: 6HBE). (**C**) *Ralstonia pickettii* NOS contains an extra heme-containing domain which might facilitate electron transfer for the enzyme (PDB ID: 3ZIY).

**Figure 4 ijms-23-10778-f004:**
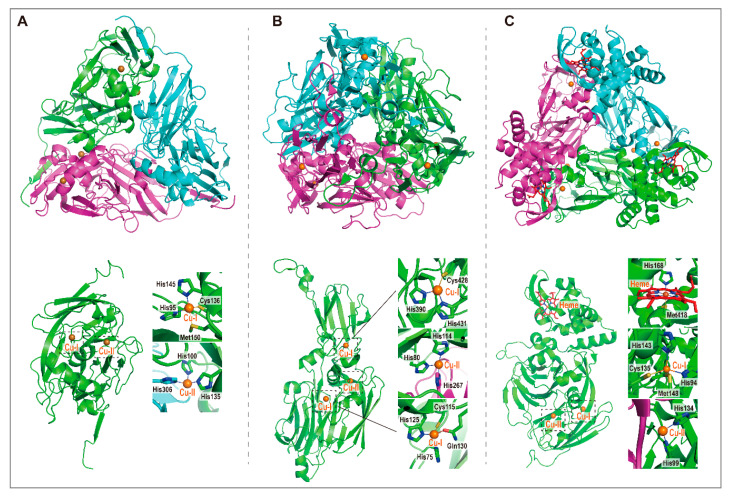
Structures of (**A**) cytochrome *cd*_1_ associated nitrite reductase (NirS) from *P. pantotrophus* and the conformational changes of the heme coordination in oxidized (**B**) and reduced (**C**) states. From oxidized state to reduced state, the coordinated residues at the distal sides of heme *c* and heme *d*_1_ changed from His17 to Met106 and from Tyr25 to solvent molecules (PDB ID: 1AOF).

**Figure 5 ijms-23-10778-f005:**
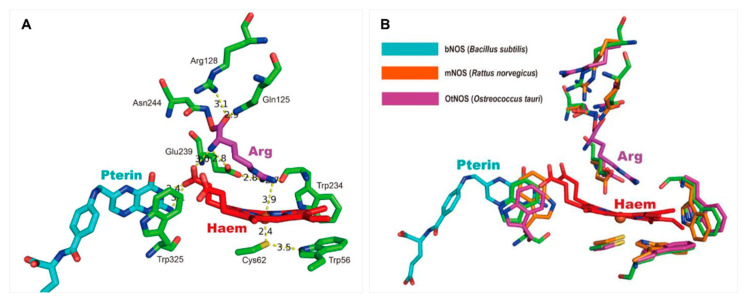
Catalytic center of bNOS. (**A**). Typical triangular relationship between heme, pterin, and substrate Arg, and the surrounding residues of Arg within the catalytic site of bNOS. The substrate Arg is fixed by a few residues via hydrogen bonds. The interaction of Trp–Cys at the proximal side of the heme has been proposed to increase the heme redox potential. Another Trp (W325) interacts with the heme propionate group and stabilizes the pterin by forming π-π stacking interactions (PDB ID: 1M7V). (**B**) Structural comparison of the oxygenase domains (catalytic sites) of bNOS (PDB: 1M7V, colored in cyan) from *B. subtilis*, mNOS from *Rattus norvegicus* (PDB: 3B3M, colored in orange) and OtNOS from *O. tauri* (predicted by AlphaFold, colored in purple). The typical triangular arrangement between heme, pterin, and substrate Arg are highly conserved among these NOSs.

**Figure 6 ijms-23-10778-f006:**
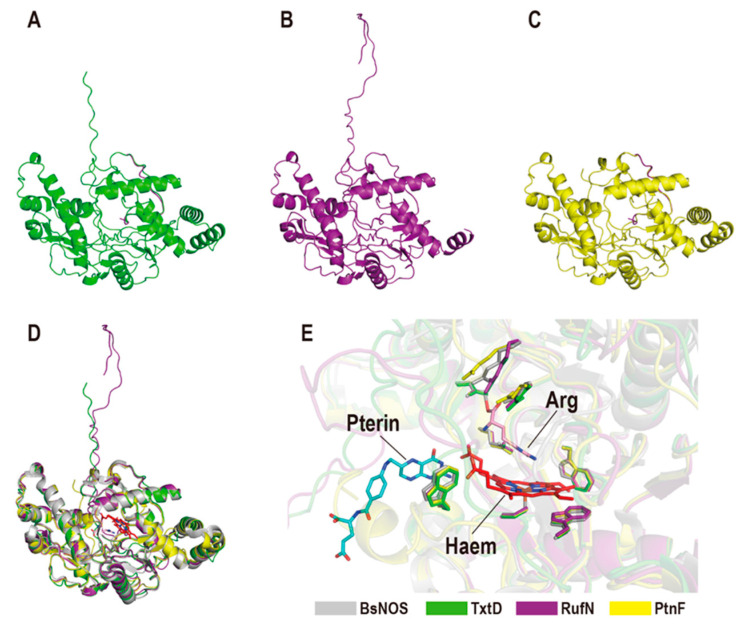
Structural comparison of the bNOSs involved in the biosynthesis of natural products with canonical bNOS. The models of (**A**) TxtD, (**B**) RufN, and (**C**) PtnF were predicted by AlphaFold and only the monomeric structure of each bNOS is used for comparison. (**D**,**E**) Despite the differences in protein sequences, all these bNOSs possess highly conserved overall structures and similar catalytic centers.

**Figure 7 ijms-23-10778-f007:**
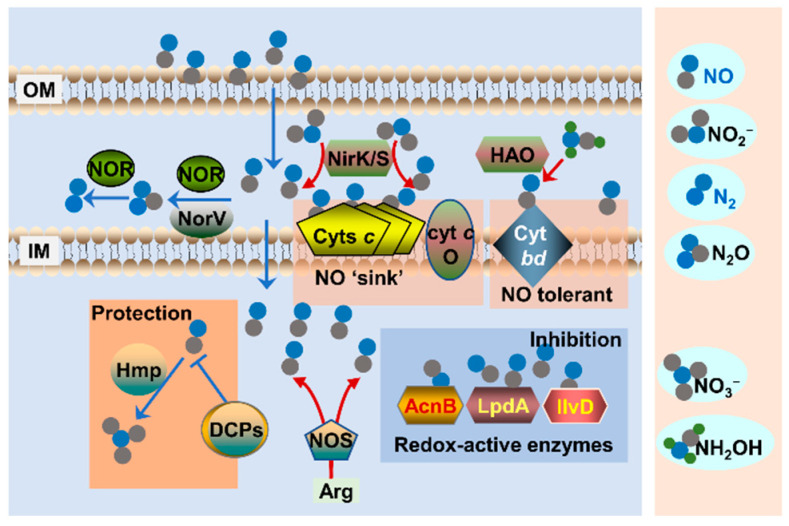
NO biology in bacteria. Bacterial cells may encounter exogenous NO, such as that released by host cells. NO can also be produced endogenously by NirK/S and HAO in the periplasm, and by NOS in the cytoplasm. NO in the periplasm can be converted to N_2_O and then N_2_. During aerobiosis, NO inhibits cyt *c* oxidase (cyt *c* O), included in cyts *c,* which interacts with NO indiscriminately and could serve as an NO sink. To grow, cells can use NO-resistant cyt *bd* to respire O_2_. In the cytoplasm, NO inhibits many redox-active enzymes, especially those carrying heme and Fe-S as cofactors. To survive and grow, a complex protection system is induced, including enzymes directly removing NO (such as Hmp) as well as damage-control proteins (DCPs).

**Figure 8 ijms-23-10778-f008:**
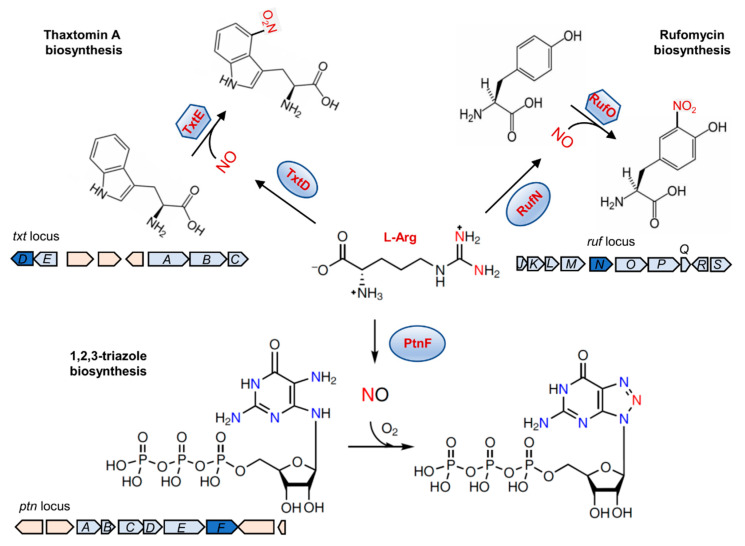
NO-dependent biosynthesis of natural products. The minimal requirements for NO-dependent biosynthesis of natural products include a dedicated NO source; TxtD, RufN, and PtnF in biosynthesis of thaxtomin A, rufomycin, and 1,2,3-triazole, respectively. In most cases, a metalloenzyme to mediate the nitration is required, such as TxtE and RufO in biosynthesis of thaxtomin A and rufomycin, respectively. Gene clusters for the biosynthetic enzymes are shown. The genes encoding the NO sources are in blue, participants required for biosynthesis are in light blue.

## Data Availability

Not applicable.

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
