# Peer review of "Nitric Oxide, Nitric Oxide Formers and Their Physiological Impacts in Bacteria"

_ijms, 2022, doi:10.3390/ijms231810778_

Round 1

Reviewer 1 Report

As a wish to the authors of the article for the following reviews, I would like to say the following. This article does not disclose the important regulatory role of the formation of DNIC iron-sulfur clusters, the association of NO with non-heme iron, and the formation of persulfides (-S-S). Perhaps the authors will be able to make an addition to these problems in the next review.

Author Response

We thank this reviewer very much for pointing out that we largely left out the important regulatory role of the formation of DNIC iron-sulfur clusters, the association of NO with non-heme iron, and the formation of persulfides (-S-S). Indeed, this is an interesting and important aspect of NO biology. We missed it because we tried to focus on NO formers, and physiological impacts of NO in bacteria. We will cover it in manuscripts in the future. 

Reviewer 2 Report

Dear author,

This paper collects and discusses in an excellent way the main and latest advances in the biosynthesis and effects of NO in bacteria. I just have a few suggestions to make to the authors.

*Although the paper deals with NO in bacteria, some reference is made to NO in humans in the introduction. That is why I consider that it would  be good if the authors also made reference in the introduction to what is known about NO in plants.

*Please indicate in the legend of Figure 1 why NirK appears with a question mark

*I suggest entering in the Figure 2 also the Nitric Oxide Synthase from Ostreococcus tauri, as representative of eukaryotic photosynthetic organism

Author Response

We thank this reviewer very much for helping us to improve the manuscript.

*Although the paper deals with NO in bacteria, some reference is made to NO in humans in the introduction. That is why I consider that it would  be good if the authors also made reference in the introduction to what is known about NO in plants.

We added this in the introduction. 

*Please indicate in the legend of Figure 1 why NirK appears with a question mark.

This process is still under debate. We explained it in the legend. 

*I suggest entering in the Figure 2 also the Nitric Oxide Synthase from Ostreococcus tauri, as representative of eukaryotic photosynthetic organism.

We revised Figure 2 to include this.